# Public Health Decision Making in the Case of the Use of a Nuclear Weapon

**DOI:** 10.3390/ijerph191912766

**Published:** 2022-10-06

**Authors:** Magdalena Długosz-Lisiecka

**Affiliations:** Institute of Applied Radiation Chemistry, Faculty of Chemistry, Lodz University of Technology, Wróblewskiego 15, 93-590 Lodz, Poland; magdalena.dlugosz@p.lodz.pl

**Keywords:** public health, nuclear weapon, radiation protection, resident management, nuke explosion simulation, radioactive contaminations, intervention activities

## Abstract

The current geopolitical situation and the war on Ukraine’s territory generate questions about the possible use of a nuclear weapon and create the need to refresh emergency protective plans for the population. Ensuring the protection of public health is a national responsibility, but the problem is of international size and global scale. Radiological or nuclear disasters need suitable decision making at the right time, which determine large effective radiation protection activities to ensure public health is protected, reduce fatalities, radiation disease, and other effects. In this study, a simulation of a single nuclear weapon detonation with an explosion yield of 0.3 and 1 Mt was applied for a hypothetical location, to indicate the required decision making and the need to trigger protocols for the protection of the population. The simulated explosion was located in a city center, in a European country, for the estimation of the size of the effects of the explosion and its consequences for public health. Based on the simulation results and knowledge obtained from historical nuclear events, practical suggestions, discussion, a review of the recommendations was conducted, exacerbated by the time constraints of a public health emergency. Making science-based decisions should encompass clear procedures with specific activities triggered immediately based on confirmed information, acquired from active or/and passive warning systems and radiometric specific analysis provided by authorized laboratories. This study has the potential to support the preparedness of decision makers in the event of a disaster or crisis-related emergency for population health management and summarizes the strengths and weaknesses of the current ability to respond.

## 1. Introduction

Since the war on Ukraine’s territory started, the key question has been how this war will end. This fact has reopened a debate on the real possibility of the use of a nuclear weapon and the risk caused by nuclear weapon detonation on the territory of any European country. In the last few months, the security circumstances in Europe have changed dramatically. Use of nuclear weapons is currently a political and military revolution in warfare and a deterrence strategy [1,2]. Explicit threats of nuclear escalation for various countries demonstrate crisis management and coercive diplomacy. The current political situation presents new challenges in the control of nuclear weapons, including prevention of the spread of nuclear weapons globally.

The idea of nuclear weapon use is the belief that they allow for defeating the enemy without the use of huge armies. Nuclear weapons use chain reaction fission of heavy element nuclei (uranium and plutonium) or a combination of fission reactions, for destructive, terroristic, or military tasks.

The World Health Organization (WHO) defines radiation emergencies as non-routine situations or events that require prompt actions related to public health and environment safety. Activities may involve reducing the radiation hazard or adverse consequences for human life, health, or property directly after detonation and in the recovery phase. 

Currently, the possibility of using nuclear weapons returns not only as media information but also in official speeches by the most important representatives of various governments. In the case of a real possibility of using nuclear weapons, it is necessary to review international recommendations and consider eventual losses and consequences for the population. Nuclear weapon application has destructive power related to various effects, including the blast, thermal radiation, direct ionizing radiation, and radioactive contamination. After World War II, nuclear weapon development was expanded and improved, especially by the US and the Soviet Union during the Cold War. However, currently, nuclear weapons are used as the object of deterrence, instead of for actual deployment in warfare [1]. Deterrence is a form of practice to sustain international security and stability in the case of a serious nuclear conflict [3]. Tactical nuclear weapons are missiles with a low power, although still exceeding the power of those that fell on Hiroshima and Nagasaki. This type of weapon is to be used “only” for a point attack on enemy targets [4], of civil or military character. 

The historical nuclear explosion in Hiroshima during World War II had an explosion yield with a strength of 15 kilotons. The estimated initial fatalities were about 70,000 people dying as a result of the initial blast, heat, and radiation effects. As a result of the bomb dropped on Nagasaki, with an explosion yield of 20 kilotons, about 40,000 people died initially; additionally, 60,000 more injured people died later. Currently, military-equipped nuclear weapons can have an explosion yield of even 1 Mt (or more) and they are even 50-times more powerful than the bomb dropped on Nagasaki. Therefore, decision making is strongly associated with the health and environmental consequences of a nuclear weapon explosion. Developing new algorithms for improvised nuclear weapon explosions helps to estimate the health and environmental effects and succor proper decision making on local, national, or global level [5,6].

The risk associated with a nuclear event resulting from the releases of radioactivity is currently much more real and requires consideration and revision of emergency response plans on local, national, and international levels for population health protection. Isotopes released from nuclear or terroristic (weapon) applications are different than from a nuclear reactor disaster. The need to reconstruct a release event and to prognose population exposure performs a role in supporting decision making and managing intervention activities, depending on the real situation. Similar modelling was developed in the case of COVID-19 disease release, emergence, and transmission [7,8]. 

Modeling of the consequences of a nuclear weapon detonation can help in the estimation of the urban infrastructure destruction, electromagnetic pulse effects, radiation protection and rescue procedures for health management, communication with affected populations, and reporting and other intervention activities needed to be conducted during a nuclear crisis action [9].

Nuclear emergency management can be divided into several phases, representing periods of time before or after the explosion. Preparedness before a nuclear event allows for establishing a set of activities, construction of emergency plan, assigning responsibilities and tasks, emergency team building, trainings, equipment tests, etc. After a nuclear event, initial immediate decisions focus on protective actions, when decisions have to be made regarding the protection of public health. The second phase, intermediate, contains the response part based on a controlled situation. Each decision made in this phase has to be based on actual measurements and collected information. Therefore, in this phase, the response has to be accurate and should contain protective actions toward humans and the environment. The third part is focused on recovery and cleaning activities for reducing radiation to acceptable levels [5,6,7].

In practice, decision strategy during the early phase after the detonation has to be designed based on the recognition of the event and estimations based on modeling (Bind, 2019). Information resulting from good quality and quantity monitoring systems confirms and substantiates the triggered protective actions for residents. The network of international (https://remap.jrc.ec.europa.eu/ (accessed on 21 August 2022)) and national early warning systems is based on monitoring stations located all over the EU Member States. An early warning system in the event of radiation emergencies performs a significant role. The European Union’s exchange of measurement data system is based on routine radiological monitoring networks. Based on local observations, decision makers receive relevant and timely information in a systematic way, in order to make decisions and take accurate actions. The real-time network of early warning system is based on the Permanent Monitoring *Stations* (PMS) (currently 37 integrated stations in Poland, 1800 stations in Germany [10], and over 5500 globally), which is dedicated to estimating the dose rate. The system stores the collected data locally on the station computer and then transmits them to the central server, located in the National Atomic Agency’s structure. The second parallel system, Aerosol Sampling *Stations* (currently 12 integrated air filter stations in Poland, thousands globally), for allows detecting and measuring the radionuclides distributed in the air and performs a role as a monitor for radioactive contaminations of the environment. PMSs contribute to the **European Radiological Data Exchange Platform** (EURDEP) network maintained by the Joint Research Centre of the European Commission. EURDEP (https://remap.jrc.ec.europa.eu/ (accessed on 21 August 2022)) **network** provides radiological monitoring results collected from automatic systems in 39 countries. Long-time measurements and a large database of collected dose rate results allow for a natural background map and radiological level visualization on a webpage. Real-time monitoring essentially reflects the natural radiation background but allows for radiological event localization at a global or regional scale. A similar role is performed by National Atmospheric Release Advisory Center (NARAC, https://narac.llnl.gov/ (accessed on 21 August 2022)) and National Data Centre (NDC) as tools for support in emergency planning and the automatic processing, including archiving and interpretation of radionuclide changes. The software allowing for modeling the releases of nuclear, radiological materials in the atmosphere allows for a real-time assessment and specific emergency response to ensure proper activities in population health crisis management. Networks detect radionuclide signals of nuclear explosions, localize the source, and allow for the reconstruction of the release and its scale. Based on the results, specific interventions should be implemented.

In this study, a simulation of nuclear weapon’s explosion allows for referring recommendations to a more “practical” situation. The modelled nuclear explosion is based on a set of assumed input data: power of the nuclear weapon, location, density of population, or wind direction and its speed. For the simulation of the nuclear explosion, NUKEMAP by Alex Wellerstein, a free online application (https://nukemap.org (accessed on 21 August 2022)), was applied. This simulator helps to show the critical region and the scale of destruction and fatalities. 

The aim of this study is to propose specific recommendations for public health protection based on the simulation results and explosion effects. The motivation behind the analysis of a simulated situation and establishing of activities that reduce the potential exposure of residents from a critical exposure region is the current political situation. In the last several months, this subject became more timely and accurate. Showing the simulation results and the scale of the possible destruction of the city and fatalities among the affected population can help in decision making and could be the best lesson for improving the emergency management to ensure effective population safety. 

## 2. Materials

NukeMap version 2.72 simulator is an online app tool based on declassified nuclear weapon effect data, developed and established by Alex Wellerstein (https://nuclearsecrecy.com/nukemap/ (accessed on 21 August 2022)). Atomic bomb impact simulator works based on 4 simple steps:

Location: The operator has to choose a place on the virtual map, where the nuclear weapon would explode. Average-sized city, in the central part with several satellite small cities located around it, was chosen for the simulation. Hypothetic agglomeration has a typical European building structure.

In this study, a highly dense agglomeration was chosen as example. Regions with a dense residence show issues and disturbances more often during an emergency evacuation or crisis situation. The number of residents reaches 1 million and the area of this agglomeration exceeds 1800 km^2^.

Power: In this approach, two scenarios were analyzed: 0.3 and 1 Mt—sample model, tactic, military equipment. 

Options to choose: type of detonation—surface—on the ground level.

NukeMap simulator allows for a visualization of direct, prompt effects, and later contamination effect. The fallout model is a scaling model that attempts to provide a general idea of the approximate distances of various levels of radioactive exposure. The model does not need detailed meteorological data about the location of the detonation, therefore, has easy structure and quick visualization option. Only a general wind speed of 15 mph was implemented.

Proposed in this study, recommendations were based on several international and national regulations, emergency working plans for reactions in crisis situations covering different areas of problems and challenges in the case of nuclear weapon use including post-accidental response and population recovery. The proposed material was based on official reports, links from government websites, the literature, or good practices in public health from various countries [11,12].

The novelty of this study is the elaboration of recommendations based on scientific knowledge and the results of the simulation. Therefore, all the stages (preparedness, early and intermediate phase, and recovery after a nuclear attack), based on up-to-date knowledge, are more appropriate for ensuring public health.

## 3. Results

To maintain public health, an appropriate response in a nuclear event emergency management should emphasize preparedness, early and intermediate phases, and recovery phase, similar to those resulting from physical, chemical, or biological hazards, including a pandemic [13]. Proper handling of each of these situations begins with building competences among people, establishing measurements of infrastructure, and anticipating consequences and their prevention based on up-to-date knowledge. 

Designing the preparedness for various interventions needs to involve citizens, public administrations, and the government, together with emergency services and management in the case of a nuclear event. The intervention level triggering a suitable action has been defined as “the level of avertable dose at which a specific protective action or remedial action is taken in an emergency exposure situation” [11,12]. 

Best practices, regulations, and guidelines link knowledge, monitoring emergency networks support multiphase decision making, and management in the real situation of a nuclear or radiological disaster. Building a ‘culture of disaster preparedness’ for citizens, communities, and emergency staff should be based on focusing on responsibility, deliberation issues, and development of effective preparedness practices. 

Deployment of evidence-based basic safety standards for public health should be based on preparedness, early and intermediate phases, and recovery phase (Figure 1). Each phase includes several recommended tasks, appropriate for a nuclear event and specific for nuclear weapon use (Table 1). In the preparedness phase, measures are taken to prepare activities, people, equipment, and procedures to ensure public health in the case of nuclear weapon use. Time phases of a nuclear accident involve prompt radiation effect, neutron activation, and atmospheric releases of radioactive isotopes. During the release, the exposure predominantly stems from the plume exposure pathways, contaminations, and inhalation exposure effects. 

## 4. Preparedness

The aim of the preparedness part is to build resources and learn about taking corrective actions after a nuclear accident. The objective of this phase is to develop short-term implications for ensuring the protection of the population potentially affected by direct radiation or radioactive contaminations (Table 1). 

Preparation of activities can be divided into several scopes in closely related areas [13,14,15].

### 4.1. Instrumental Equipment

-Building and routinely using online monitoring systems for searching for artificial radionuclides in the air or controlling the changes in the dose rate levels.-Active devices/monitors for contamination or radionuclide control (in situ equipment in the forms of radiometers, spectrometry systems, or counters) calibrated in certified laboratories.-Building and establishing a highly professional laboratory service for monitoring air, water, and food contaminations using radiochemical separation methods, based on certified or reference materials used in proficiency tests and interlaboratory comparisons, accredited or authorized by National Atomic Agency.-Dosimetry devices to protect emergency workers and the public.-Decision support systems—software for the simulation of the regional and global situation, radionuclide, and dose distribution in time intervals.-Respiratory system protection.

### 4.2. People

-Building competences, new skills, creation of the management structure, gathering professionals and experts, decision makers, representatives of government and local administrations, authorities, coordinators, medical staff, etc.-Building an emergency team structure for early response, organizing courses, trainings, workshops for better understanding of the radiation protection rules, personal equipment, self-sheltering, etc.-Maintaining the population.-Teaching, raising awareness, and sharing knowledge about radiation and the principles of radiological protection among students of physics, chemistry, or related studies.

### 4.3. Instruction and Procedure Development and Establishment

-Nuclear or radiation emergency plan constantly updated and tested.-Developing and testing procedures and national or regional instructions of intervention actions.-Establishing information exchange channels between staff on various decision-making levels and organization structures.-Stable iodine distribution protocols (if it is applicable) for the population.-Population communication protocols, including modern systems (apps for mobile phones) and common sources of information, etc.-Resource management: instruction for radiometric food and water analysis, protocols for clean water supply, food distribution, etc.

### 4.4. Medical Care Infrastructure and Sheltering Overview

-Establishment of temporary hospitals.-Refurbishment of shelters for massively exposed residents.

International cooperation and exchanging experience and monitoring systems for warning and triggering emergency response serve to promote and enhance safety globally, alongside training courses, exercises, testing of the radiometric equipment, and proper calibration procedure for spectrometers or dosimeters. Laboratory preparation of methods or procedures for a good-quality radiochemical analysis should participate in interlaboratory proficiency tests and intercomparisons. 

Part of preparedness for an eventual nuclear event is building a structure of radiation protection experts, professionals, environmental protection workers for cooperation, and good practice elaboration. Field exercises for fire brigades or environmental protection personnel specializing in working with ionizing radiation should also be carried out on a regular and competitive basis [16]. 

Council Directive 2013/59/Euratom of 5 December 2013 and International Commission on Radiological Protection [17,18] lay down basic safety standards and regulations for protection against the dangers arising from exposure to ionizing radiation. Decision making and management after a nuclear event should be based on three principles: justification, optimalisation, and the use of dose reference levels [18]. Based on this, each European country, voivodship, and radiation-using facility should have to constitute a radiation emergency management system, including planning, responsibilities, emergency staff, and direct instructions, etc. Well-developed emergency plans, procedures, and instructions are essential and must be implemented and adjusted for organization of suitable crisis management. In the case of radiation exposure, detailed instructions for residents are needed to ensure sheltering and respiratory tract and skin protection to avoid internal and external contamination. 

## 5. Early and Intermediate Phases

For accurate and specific decision making and proposals for future planning, this study presents a simulation of a 1 Mt nuclear weapon explosion for a chosen city. In this phase, the following effects occur: the blast, thermal radiation, and prompt ionizing radiation; they cannot be reduced or limited, because of the immediate physical, chemical, and radiation processes and their direct influence on the bodies [11,14]. 

The use of a single nuclear weapon can generate several effects divided into two parts: 

### 5.1. Surface Nuclear Burst Effects

Estimation of the direct detonation effects of nuclear weapon use is difficult and depends on various parameters, its power, and the place of detonation (under the ground, on the surface or in the atmosphere). A nuclear attack is highly destructive. The most significant and harmful effects are most likely to occur in the case of surface or near-the-ground detonation due to the physical, thermal, and radiation contamination, where the detonation power destroys matter, civil residential buildings, military objects, and communication infrastructure at a large distance. For a 1 Mt power weapon detonated at the ground level, the crater has about 90 m depth and can reach 400 m radius. In the case of detonation in an urban area with a high density of population (a large agglomeration) and building structure, the destruction and fatalities are massive. For airburst at 1 km, a high fireball and direct radiation effects are lower; however, blast damage, thermal radiation, and light blast have a large distance (Table 2).

The nuclear weapon detonation with yield 0.3 Mt generates a lower scale of effects (Table 3) for ground and airburst types. 

Organic matter in the region of detonation will vaporize in a fireball. The simulation predicted 100% fatalities over a 2–3 km radius from the detonation point. Due to the prompt radiation effect at the level of 5 Sv, additional fatalities are expended over the next 30 days. The destruction area can reach over 400 km^2^—which is equal to an average agglomeration with a civil residential structure. However, next, the radioactive fallout must be taken into consideration. An additional dose rate for the population resulting from radionuclide distribution and internal and external contamination is highly correlated with the meteorological condition of the atmosphere and decision making early after the nuclear event. 

### 5.2. Radioactive Fallout 

Based on the simulated 1 Mt and 0.3 Mt nuclear weapon detonation results (Table 2, Table 3 and Table 4), several dangerous effects were observed. Modeling was performed for a high-density residential agglomeration city with several satellite villages, with a total of 1 million citizens. Directly as a result of the nuclear explosion on the ground, over 350,000 fatalities and over 250,000 injured residents were estimated. For airburst of nuclear detonation at 200 m, over 395,000 fatalities and 376,000 injuries were estimated. Over 800,000 deaths and 1,000,000 people could be injured in the light blast damage region, as a result of building destruction and collapse, class breaking, and thermal skin burns, respectively, for ground-level and airburst-type attacks (NukeMap simulator results). 

Using a 300 kton nuclear weapon generates over 177,000 fatalities but the estimated number of injuries is similar, reaching 249,000 residents. A considerably higher percentage of the rural population would survive. In this approach, scaling models were applied for approximate distances of various levels of radioactive exposure, but without using specific weather conditions. This approach does not use type of terrain, the ratio of fission to fusion reactions in the bomb, or activated fallout in the dose estimation. Therefore, modelling is simple and does not need much specific information about localization. 

Responsible for the coordination and implementation of national emergency procedures is the government in cooperation with national Atomic and City Hall administration, together with national and voivodeship administration departments of crisis management. In the early phase, identification, notification, and activation should be triggered [19].

Historical nuclear bomb detonation releases several kilograms of radioactive materials: uranium in the case of “Little Boy” and plutonium in the case of “Fat Man”. In the case of a hypothetic nuclear reactor explosion, even several tons of radioactive material could be released as a mix of nuclear fuel and fission products, including gaseous elements. The amount of radioactive material released and its quality are critical for further action and decisions. Active management in the early period of a nuclear weapon containing ^239,240^Pu in the form of PuO_2_ or ^235,238^U isotopes in the form of UO_2_ aimed at reducing the inhalation intake performs a very important role in mitigating the radiological consequences. Plutonium and uranium are highly radiotoxic alpha emitters. For the alpha particles, the critical pathway of intake is inhalation. Talaat and Baldez [20] noted elevated doses in the trachea and lungs only, since the average free path for alphas in the tissue is very short, less than 50 microns in matter, including tissue. Therefore, reducing the internal exposure by applying respiratory protection systems is necessary. 

Inhalation of aerosols after the detonation generates a rapid stage of plutonium or uranium isotope accumulation in the body and slow radionuclide elimination from the body and a gradual increase in the absorbed dose over time. Inhalation intake of uranium oxides of about 400 kBq generates a lethal dose (effective dose coefficient, inhalation type S, equal to: 1.2 × 10^−5^ Sv/Bq [21]), leading to fatal cases during the month after an accident as a result of acute interstitial pneumonitis. An intake in a range from 40 to 400 kBq generates serious deterministic effects, mostly pneumosclerosis and stochastic effects in the form of cancer of the lungs in later years. 

A nuclear reactor disaster and nuclear weapon use yield rather different effects. In the case of a nuclear bomb detonation, uranium and plutonium compounds are highly toxic and dangerous, due to the radioactive isotopes’ decay emitting high-energy alpha particles. For this reason, limiting internal exposure performs a significant role in population health and personal protective equipment should be implemented as a part of the requirements. 

After the explosion, fission products released in the atmosphere attach to airborne particles of varying sizes; however, fine-sized aerosols generate the highest inhalation hazard [22,23,24]. Radiation monitoring should involve analysis of gaseous and aerosol fractions and should include alpha, beta, and gamma radiation for control of all radiation sources. 

Due to direct radiation exposure and radioactive fallout, time optimalization has a significant meaning. Decision making is necessary when the situation becomes untypical or changes dynamically; therefore, within a few hours after the nuclear bomb explosion, the problem is recognized and decision making is based on estimations or analysis of collected material in laboratories on dedicated alpha spectrometry systems for dose rate estimation. Dose rate or effective dose results can trigger several activities (Table 5).

A set of early intervention activities, evacuation, relocation, and water and food consumption limitations, can be primarily implemented based on the real existing exposure. The level of exposure (dose rate, effective dose, radionuclide concentrations, etc.) determines the type of implemented intervention for the region and the group of residents (Table 5). Only radiometric measurements acquired in an emergency mode in situ or on samples collected on the site can provide valuable, robust information, which can trigger specific interventions and activities. Only suitable coordination of work between radiometric experts, radiometrists, chemists, crisis management staff, and administration representants can allow optimalization of the time and shortening of the real exposure time of the burdened population. Establishing information exchange channels is a basic part of the preparedness phase and to avoid chaos in information streaming, this task should be arranged and corrected during every training activity.

Due to a large region and a high number of residents, reliable and accurate information should be quickly delivered to residents. Citizens should be informed about the threat via alarm sirens, RSO applications, local TV, radio, and internet portals [26,27]. There will also be advisory procedures and relaying of information on needed actions for residents in a given situation in the district. Crisis management, emergency staff, and volunteers in each district will organize action on site. The consequences are generally linked with the early decision-making process undertaken after the actions [11,12]. 

A formal registry of exposed and injured residents should be established for proper management of evacuation and establishment of temporary hospitals or medical care points. To coordinate such various activities and minimalize the number of victims and the obtained dose by survivors, suitable preparedness in time of peace is highly recommended. Related to nuclear emergency preparedness, emergency team courses, review or refreshment of instructions, protocols, and procedures for various provinces and districts must be ready.

After World War II, many concepts and regulations were created regarding civil defense and building security against hostile forces in the biggest city agglomeration. For example, proper layout of rooms, including equipping them with installations and internal devices that will ensure appropriate conditions for people hiding in them to stay there for at least 3 days, were implemented. In most cities, about 600 points of shelter and temporary hiding places for citizens in total for all five districts of the agglomerations were established. Most of the points are located in regions of direct surface building damage, building collapse, and high dose rate or contamination level. Access to such underground structures may be difficult or impossible in high-density building structures in the city center. Therefore, some shelter structures are located underground, beneath local parks or green urban areas. 

In the early and intermediate phases, the designation of an emergency area (zone) with a dose rate limit equal to 100 mSv/h with a prohibited entrance for the public should be implemented. According to the simulation, this highly contaminated area covered about 18 km^2^ and its localization is correlated with local meteorological conditions at the explosion time and the direct missile detonation point. 

Taking into consideration the emergency area with a dose rate exceeding 100 mSv/h, other highly contaminated regions with high variability in radiation should be indicated, including an evaluation area, temporary evacuation area, and an order to stay in a building with sealed windows or in underground shelters. Designation of all these areas needs to be conducted based on a direct dosimetry analysis, based on active monitors used in situ. An alternative approach can be used based on the simulator real-time on-line decision support system (RODOS) application [28,29]. The simulator is dedicated to nuclear emergency management at national levels. The European Commission’s Radiation Protection Research Action, based on own regulations and historical nuclear accidents, developed a decision support system. Based on methodological conditions, developed models and databases simulate the release and radioactive fallout in the vicinity of the accident point. The simulation estimates, analyzes, and estimates consequences and suitable protective actions for various nuclear events. In the early phase, dose and contamination monitoring in real time should be estimated for various distances from the detonation point. To ensure proper radiation protection in the early emergency response, sheltering and evacuation should be managed. Relocation is intended to minimize the health risks of external and/or internal contamination and, in practice, to minimalize radiation exposure and serious health risks [30]. 

Next to the simulation prognoses, the exposure dose of the residents and staff should be carefully monitored and controlled in real time. 

Due to the necessity to coordinate many activities, there is a high risk of information and organizational chaos at the same time. Therefore, one of the basic conclusions in risk assessment should be the emphasis on good cooperation of services and the development of communication channels and decision-making procedures. In order to shorten the time of activities and increase their efficiency, it is necessary to conduct training and practical exercises, at the local, provincial, national, and international level. 

The intermediate phase covered all protective actions, including providing food, medical, and psychological help, to the surviving residents and ensuring epidemiological safety for the city and the region. In this phase, all activities should be conducted based on confirmed and robust information about the radiometric situation, dose rate in ambient air, and contamination of the surface.

During the early and intermediate phases, many of the fatally injured residents would die immediately or weeks or a month later; some of them would need professional medical help and would have to be hospitalized. Care for the injured would place a heavy burden on medical staff and professionals. 

After the previous phases end, the recovery and cleaning phase begins and should be a long-term one, focused on continued support of residents and reductions in the levels of radiation in the environment so that they can be considered acceptable.

## 6. Recovery Phase

The aim of the phase (Table 1) is to support the affected populations in the social, financial, and economic confusion caused by the nuclear detonation.

In this phase, cleaning the terrain and management of radioactive waste are essential parts of the reduction in the dose rate for the population and settle on recovery of the urban structure of terrain. Long-term health screening and total dose estimation for emergency staff and residents are the general parts of the recovery phase. However, long-term relocation influences various sociological, mental, and health issues. Resettlement of a large number of people to new living places causes various difficulties. Evacuees would have lost their jobs or faced the loss of a loved one among their family or friends, have to be provided with new living conditions, with most of them separated from their families and friends at a large distance. At a greater scale, evacuees have to rebuild a society or become part of the community. After cleaning and infrastructure recovery, in some regions, residents may be allowed to return and rebuild or renovate new housing estates. Therefore, for some evacuees, it is possible to return to their own homes. Adaptation to new conditions may intensify stress and increase the risk of mental illness and neurosis, affect fertility, human behavior, and relationships with family members. Moreover, the life expectancy of the exposed population would be shortened, in a range from 4 to 5 years [25]. Due to radiological exposure, some male survivors would be temporarily sterile. Therefore, a reduction in fertility is possible. Those residents of childbearing age exposed during an explosion may show genetic injury, causing cancer or genetic defects to appear in the next generations. After a few months or years, radiation exposure effects can be observed. 

Continual radiometric analysis (dose assessment, monitoring, fallout) should be conducted in this stage to ensure radiological protection and support epidemiology analysis. Personal dosimeters for workers or even the general population linked with mobile applications are a good option.

After a nuclear bomb detonation, water and food distribution should be coordinated by the government. Water and food produced locally need to be controlled via a radiometric analysis to prevent gastric intake of increased radionuclide concentration due to contaminated food or water consumption [31,32]. Active support from the government, local state structures, and experts is necessary to reconstruct and rebuild conditions to live [33] and support for affected populations.

Active programs to enhance recovery after a nuclear attack should be regulated in several subjects: life support, epidemics and diseases, late and genetic radiation effects, economic breakdown, and contamination monitoring in the environment [30]. The recovery phase includes radionuclide monitoring in environmental, decontamination, food, and water control. Radiometric analysis covers estimating the radiation dose in the environment. Currently, the population is much more aware of radiation protection procedures and selfcare rules, especially after the COVID-19 disease experience [34,35]. 

The recovery dynamic would be correlated with the political and economic situation at a national and international level. Social and political organizations and the restoration of a functioning economy influence the human sense of stability. Therefore, the recovery phase can span even tens of years after a nuclear attack to ensure conditions in which people can be healthy [30,33,36].

Due to high fatalities in the early and intermediate phases, a general disruption to public health organizations is possible. Temporary loss of medical staff could reduce the quality of the public health practices and disease surveillance systems. In the recovery phase, these issues should be resolved and organized for new challenges in the epidemiological research and care for people affected by the radiation exposure.

## 7. Discussion

Many medial topics about potential nuclear aggression appear in mass media; therefore, it is interesting to answer the question of what would happen and how safe a population would be if a nuclear detonation took place now. Nuclear weapon use would be enormously destructive and would lead to decision making regarding the protection of public health, reductions in harm, and social-protective actions. Currently, people are thinking about actually using nuclear weapons for the first time since 1945. Therefore, it is necessary to review emergency protocols and establish suitable recommendations regarding preparedness and health surveillance of populations potentially affected by nuclear weapon use.

## 8. Conclusions

Nuclear weapon use with an explosion yield of 0.3 and 1 Mt was applied to establish the required decision making for a population. There is a need to refresh the protocols for the protection of public health in most European countries, containing clear procedures with specific activities triggered immediately based on confirmed information. Active or/and passive warning systems and radiometric-specific analysis provided by authorized laboratories have an essential role in proper decision making. Science-based decisions can help support preparedness in an emergency action for population health management. 

## Figures and Tables

**Figure 1 ijerph-19-12766-f001:**
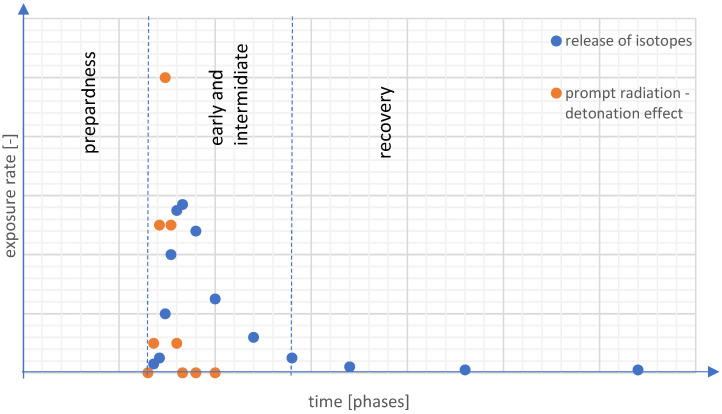
Exposure rate (biological effects) in phases before and after nuclear weapon use (illustrative diagram).

**Table 1 ijerph-19-12766-t001:** Phases and recommendations for the decision-making process.

Preparedness	Early and Intermediate Phases	Recovery
online monitoring systems for the support of decision making nuclear or radiation emergency plans continuous review and training developing and testing procedures and national and regional instructions passive infrastructure for radiation protection—shelters active devices/monitors for contamination identification, modern systems with low limits of detection	in the first minutes and hours collecting information needed for decision making, based on monitoring systems for the emergency response radiometric measurements in situ and dose estimations for emergency staff, workers, and the public communication with residents and indication of shelters and hospitals, medical centers distribution of breathing apparatuses and respiratory masks for self-protection establishment of a roster of victims, injured, or lost residents, organization of evacuation and relocation, coordinating and managing establishing a zone with prohibited entrance providing water, food, medical and psychological help to the survivors ensuring epidemiological safety for the city and the region, other protective actions toward people and animals	long term medical and psychological care for the population regular monitoring of radionuclide concentration in food and water supply monitoring of determined and stochastic radiation effects, epidemiological approach recovery of the urban structure of the city or agglomeration outside the zone, radiometric and dosimetric systems in the environment

**Table 2 ijerph-19-12766-t002:** A nuclear weapon (1 Mt) simulation (first scenario) of effects based on the NukeMap simulator.

Effect	Radius [km]		Surface [km^2^]		Destruction Effects
	Ground-Level Detonation	Airburst, 1 km High	Ground-Level Detonation	Airburst, 1 km High	
Nuclear fireball	1.26	0.97	4.96	2.93	Vaporization of matter
Radiation	1.94	1.66	11.8	8.67	5 Sv dose (fatal in about 1 month)
Heavy blast damage	2.18	2.39	14.9	18.0	Heavy damage, fatalities at 100%
Blast damage	4.58	5.56	65.8	97.2	Residential damage, building collapse
Thermal radiation	10.7	12.5	359	494	Third degree burns on the skin
Light blast	11.8	14.5	435	662	Glass window break

**Table 3 ijerph-19-12766-t003:** A nuclear weapon (0.3 Mt) simulation (second scenario) of effects based on the NukeMap simulator.

Effect	Radius [km]		Surface [km^2^]		Destruction Effects
	Ground-Level Detonation	Airburst, 1 km High	Ground-Level Detonation	Airburst, 1 km High	
Nuclear fireball	0.78	0.60	1.89	1.12	Vaporization of matter
Radiation	1.61	1.26	6.7	4.99	5 Sv dose (fatal in about 1 month)
Heavy blast damage	1.46	1.76	8.1	9.75	Heavy damage, fatalities at 100%
Blast damage	3.0	4.0	29.5	50.9	Residential damage, building collapse
Thermal radiation	6.3	7.4	126	172	Third degree burns on the skin
Light blast	7.9	10.7	195	356	Glass window break

**Table 4 ijerph-19-12766-t004:** Radioactive fallout based on the NukeMap simulator for 0.3 and 1 Mt.

Fallout Contour with Min Dose Rate	Size (Width [km]/Downwind Cloud Distance [km]/Area [km^2^]) for 0.3 Mtons	Size (Width [km]/Downwind Cloud Distance [km]/Area [km^2^]) for 1 Mtons
0.01 Sv/h	46.3/262/10,100	102/416/33,900
0.1 Sv/h	30.1/181/4710	72.2/308/18,000
1.0 Sv/h	13.8/101/1350	42.5/201/7100
10 Sv/h	2.32/8.81/27	12.8/92.8/1140

**Table 5 ijerph-19-12766-t005:** Intervention levels (IL) of avertable dose for the population [25].

Primary Protective Actions	Poland	Ireland [26]	Denmark	Sweden	ICRP [18]	IAEA 2014 [11]	EPA 2017
Evacuation	100 mSv/7 days	100 mSv/7 days	70 mSv/7 days 10 mSv/day in max. 1 week	3–30 mSv/day	50–500 mSv/<1 week	50 mSv < 1 week	10 to 50 mSv/projected dose over four days
Sheltering	10 mSv/2 days	50 mSv/7 days	10 mSv	1–10 mSv/day	5–50 mSv/<1 day	10 mSv/< 2 days	10 to 50 mSv/projected dose over four days
Temporary relocation	30 mSv/30 days	100 mSv in first year	10 mSv/month or 1 Sv life dose	5–50 mSv in the first month	5–15 mSv life dose	30 mSv in the first month	
Permanent resettlement	1 Sv/50 years (adult) or 70 years (kids) Or 10 mSv/2 years					1 Sv life dose	
Water and food controls	Based on national regulations	1 mSv first year					

## Data Availability

Data available on request from the corresponding author.

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
