# Peer review of "Public Health Decision Making in the Case of the Use of a Nuclear Weapon"

_ijerph, 2022, doi:10.3390/ijerph191912766_

Round 1
Reviewer 1 Report
Very interesting attempt to recall how terrible is possibility of nuclear attack. I am afraid that appearing in right time. Some comments came to me about the assumed scenario, which seems to me the weakest point of paper. I suggest of taking into consideration at least in a table various scenarios.
1. The 1 Mt device must be a thermonuclear, so here you should include also fusion besides fission. By the way, I think that typical yield is lower, in range of 250 kt-500 kt TNT equivalent. Moreover, what is called a “tactical nuclear weapon” is between 0.3 kt to 170 kt. So maybe it would be interesting to have a comparison of ranges on which different factor acts for a few cases of different yield, in range from 0.3 kt to 1 Mt.
2. I am not sure if the assumed height of explosion (on ground) is the most possible one. Perhaps it would be interesting to discuss differences between on ground and at elevation of 1 km explosion, both WW2 devices were detonated above ground to increase the devastating effects.
I do hope that all paper will remains only academic.
Author Response
The author would like to thank Reviewers for their time and for all suggestions and comments.
I believe that all needed corrections and additional descriptions increase quality of the article and dispel doubts connected with eventual publication. All response/corrections have been colored red in the text of the manuscript and under the details comments of the Referee.
Reviewer 1
Very interesting attempt to recall how terrible is possibility of nuclear attack. I am afraid that appearing in right time. Some comments came to me about the assumed scenario, which seems to me the weakest point of paper. I suggest of taking into consideration at least in a table various scenarios.
- The 1 Mt device must be a thermonuclear, so here you should include also fusion besides fission. By the way, I think that typical yield is lower, in range of 250 kt-500 kt TNT equivalent. Moreover, what is called a “tactical nuclear weapon” is between 0.3 kt to 170 kt. So maybe it would be interesting to have a comparison of ranges on which different factor acts for a few cases of different yield, in range from 0.3 kt to 1 Mt.
As Reviewer suggested 0.3 and 1 Mt yielded nuclear weapon have been consider.
- I am not sure if the assumed height of explosion (on ground) is the most possible one. Perhaps it would be interesting to discuss differences between on ground and at elevation of 1 km explosion, both WW2 devices were detonated above ground to increase the devastating effects.
Yes, two airburst on ground-level and at 1000 m high have been analyzed.
I do hope that all paper will remains only academic. Yes, this manuscript has only academic character.
Reviewer 2 Report
Thank you for submitting this article. I have a few minor comments that you should consider.
* Figure 1: the vertical axis has no units or values. How does the exposure rate compare to biological effects to humans?
* Section 3.1/2/3 and Figure 2: the list format is strange and hard to read. What are you trying to convey to the reader? This is not a very good way to present data. The formats change from one section to another. Please consider another option.
* Table 2: consider a graphic rather than a table for this data. The data doesn't make sense as-is. The 10Sv/hr area is 12.8km wide, 92.8km long, but has an area of only 1.14km^2?
* Fallout: page 4 says that the simulation was run with a ground-level blast but your fallout section (page 8-9) only talks about nuclear material dose consequences. Surface explosions through up a huge amount of soil and building materials as activated fallout. They are far "dirtier" than higher altitude detonations. They are also less damaging to infrastructure, so they are not preferred from the attacker's point of view. It is not clear if the simulation included activated fallout in the dose estimates or not. Either way, you should explain what was done and what was included in the results.
* Table 3 is hard to read in this format.
* Section 3.6 seems weak. It doesn't address the cleanup and remediation of the effected areas. I don't see much in this section that is specific to a nuclear detonation vs. a generic disaster.
Author Response
Reviewer 2
Thank you for submitting this article. I have a few minor comments that you should consider.
* Figure 1: the vertical axis has no units or values. How does the exposure rate compare to biological effects to humans?
This suggestion has been applied.
* Section 3.1/2/3 and Figure 2: the list format is strange and hard to read. What are you trying to convey to the reader? This is not a very good way to present data. The formats change from one section to another. Please consider another option.
Graphical form has been changed on table.
* Table 2: consider a graphic rather than a table for this data. The data doesn't make sense as-is. The 10Sv/hr area is 12.8km wide, 92.8km long, but has an area of only 1.14km^2?
Yes, there was a mistake in notation (dot/comma). It has been corrected.
* Fallout: page 4 says that the simulation was run with a ground-level blast but your fallout section (page 8-9) only talks about nuclear material dose consequences. Surface explosions through up a huge amount of soil and building materials as activated fallout. They are far "dirtier" than higher altitude detonations. They are also less damaging to infrastructure, so they are not preferred from the attacker's point of view. It is not clear if the simulation included activated fallout in the dose estimates or not. Either way, you should explain what was done and what was included in the results.
It is very difficult to accurately model radioactive fallout. There are many relevant variables: the height of the blast detonation, the ratio of fission to fusion reactions in the bomb (most thermonuclear weapons derived at least 50% of their yield from fission reactions), type of terrain (forest/buildings/ rural). Therefore I decided to put in the manuscript 4 scenarios: detonations on the ground-level and at 1km high in the air and I simulate two additional detentions yielded 0.3 and 1 Mt of nuclear weapon, to show differences in the scale of effects.
* Table 3 is hard to read in this format.
Table has form of review for intervention levels and intervention actions, based on general IAEA or ICRP or national (European countries) regulations. I decided to keep the information in tabular form, for easy of reading. In graphical form it would be much more hard to understand.
* Section 3.6 seems weak. It doesn't address the cleanup and remediation of the effected areas. I don't see much in this section that is specific to a nuclear detonation vs. a generic disaster.
Some additional information about recovery phase has been added.